# Nitrate, Nitrite, and Iodine Concentrations in Commercial Edible Algae: An Observational Study

**DOI:** 10.3390/foods13162615

**Published:** 2024-08-21

**Authors:** Patricia Casas-Agustench, Jade M. Hayter, Odelia S. B. Ng, Lauren V. Hallewell, Nathaniel J. Clark, Raul Bescos

**Affiliations:** School of Health Professions, Faculty of Health, University of Plymouth, Plymouth PL4 6AB, UK; hayterjade94@gmail.com (J.M.H.); odeliang.ob2001@gmail.com (O.S.B.N.); nathaniel.clark@plymouth.ac.uk (N.J.C.); raul.bescos@plymouth.ac.uk (R.B.)

**Keywords:** edible algae, nitrate, nitrite, iodine

## Abstract

Edible algae are a natural source of nutrients, including iodine, and can also contain nitrogen in the form of nitrate (NO_3_^−^) and nitrite (NO_2_^−^) as they can fix nitrogen from seawater. This study aimed to analyse the NO_3_^−^, NO_2_^−^, and iodine concentrations in eighteen macroalgae and five microalgae species commercially available in the United Kingdom. NO_3_^−^ and NO_2_^−^ concentrations were measured using high-performance liquid chromatography (HPLC), and iodine was determined using inductively coupled plasma mass spectrometry (ICP-MS). NO_3_^−^ and iodine concentrations in macroalgae (NO_3_^−^: 4050.13 ± 1925.01 mg/kg; iodine: 1925.01 ± 1455.80 mg/kg) were significantly higher than in microalgae species (NO_3_^−^: 55.73 ± 93.69 mg/kg; iodine: 17.61 ± 34.87 mg/kg; *p* < 0.001 for both). In the macroalgae group, nori had the highest NO_3_^−^ (17,191.33 ± 980.89 mg/kg) and NO_2_^−^ (3.64 ± 2.38 mg/kg) content, as well as the highest iodine content. Among microalgae, *Dunaliella salina* had the highest concentration of NO_3_^−^ (223.00 ± 21.93 mg/kg) and iodine (79.97 ± 0.76 mg/kg), while *Spirulina* had the highest concentration of NO_2_^−^ (7.02 ± 0.13 mg/kg). These results indicate that commercially available edible algae, particularly macroalgae species, could be a relevant dietary source of NO_3_^−^ and iodine.

## 1. Introduction

Edible algae are photosynthetic aquatic organisms widely used in cuisines around the world for their nutritional value, which includes vitamins, minerals, antioxidants, and essential fatty acids [1]. These algae can be consumed as food or dietary supplements. They can be classified into macroalgae and microalgae. Macroalgae, commonly known as seaweeds, are multicellular algae and include examples such as kelp, nori, and dulse [2]. Microalgae are microscopic, unicellular, or multicellular algae, with *Spirulina* and *Chlorella* being notable examples [3].

This study focused on the nitrate (NO_3_^−^) content in edible algae as this anion is increasingly being considered a conditionally essential nutrient [4,5,6]. Dietary NO_3_^−^ can enhance nitric oxide bioavailability, which is a key signaling molecule involved in many physiological processes, including cardiovascular regulation, neuronal signaling, and immune responses [4]. Additionally, dietary NO_3_^−^ from green leafy vegetables or beetroot juice has a prebiotic effect on the oral microbiome, increasing the abundance of bacteria that can reduce the risk of periodontal disease [7,8,9,10]. This new evidence is shifting the traditional view on dietary NO_3_^−^ as a potential carcinogen [11]. Although an Acceptable Daily Intake (ADI) of 3.7 mg/kg/day of NO_3_^−^ is still retained [12], the consumption of NO_3_^−^-rich foods such as rocket, spinach, lettuce or beetroot can exceed the ADI levels [13]. However, recent evidence suggests a protective effect of plant-based NO_3_^−^ against cancer and cardiovascular disease [14].

Current knowledge about the NO_3_^−^ content in commercial edible algae is limited. A recent study by Martin-Leon et al. (2021) [15] reported substantial variation among different commercial algae species in Spain, with nori and kombu showing the highest concentrations of NO_3_^−^ (±3000 mg/kg). However, the analysis method used in this study lacked the sensitivity to measure NO_3_^−^ at levels below 500 mg/kg. To address this limitation, the current study employed high-performance liquid chromatography (HPLC) with high sensitivity (up to 0.1 pmol) for the measurement of NO_3_^−^ and nitrite (NO_2_^−^) levels [16].

When NO_3_^−^ is taken up by edible algae, it can be reduced to NO_2_^−^, which is then transported to the chloroplasts for reduction to ammonium [17]. Nitrogen incorporation is often rate-limited by the control of ammonium assimilation [17]. This is relevant because while NO_3_^−^ is safe even at high doses, NO_2_^−^ can cause serious harm at considerably lower levels [18]. Consequently, an ADI level of 0.07 mg/kg body weight/day of NO_2_^−^ was established by the European Food Safety Authority (EFSA) [19]. To the best of our knowledge, no previous study has investigated the NO_2_^−^ concentration in commercial edible algae. This represents a significant gap in the literature that warrants further research. 

In addition to NO_3_^−^ and NO_2_^−^, edible algae can also serve as an important source of iodine. This is relevant because iodine can potentially interfere with NO_3_^−^ uptake and its subsequent reduction to NO_2_^−^ [20]. Moreover, iodine is essential for thyroid hormone synthesis, but an excess of iodine intake can also lead to thyroid gland dysfunction and goitre [21]. The recommended intake of iodine is 150 μg/day for adults, with the tolerable upper intake level (UL) set at 600 μg/day by the Scientific Committee on Food (SCF) [22]. Following this, some manufacturers of edible algae recommend a maximum intake of 5 g/day to reduce the risk of iodine toxicity. However, most commercial edible algae do not provide information about their iodine concentration on the packaging. Current scientific data shows significant variation in the iodine concentration of commercial edible algae, ranging from 30 to 31,000 μg/g [23]. 

Therefore, the main goal of this study was to analyse the NO_3_^−^, NO_2_^−^, and iodine concentrations in commercial macroalgae and microalgae available in the United Kingdom (UK). A secondary aim was to compare the NO_3_^−^ and NO_2_^−^ levels in these edible algae with the current ADI and UL levels for NO_3_^−^, NO_2_^−^, and iodine, respectively. We hypothesised that certain edible algae could contain substantial amounts of NO_3_^−^, potentially surpassing the levels found in terrestrial green leafy vegetables like rocket (6000–7000 mg/kg) or beetroot (3000–4000 mg/kg). Additionally, we also hypothesised that consuming standard portion sizes of edible algae (5 g/day) would not result in NO_3_^−^, NO_2_^−^, and iodine levels exceeding the ADI and UL thresholds.

## 2. Materials and Methods

The concentrations of NO_3_^−^, NO_2_^−^, and iodine in 23 commercially available edible algae in the UK (Table 1) were measured. These products were purchased in July 2022 and stored at −20 °C pending analyses. 

### 2.1. Preparation of Edible Algae Extracts

The extraction of NO_3_^−^ and NO_2_^−^ from edible algae was performed according to Pinto et al. (2015) [24] with some modifications. Five grams of frozen edible algae were pulverised using a grinder (Krups F20342 Grinder, Krups, Germany), except for those edible algae already in powder form. Then, 1.5 g of pulverised/homogenised or powdered edible algae was diluted with 100 mL of hot (70–80 °C) ultrapure water (Purelab OptionQ, Oxford, UK) in a 125 mL conical flask. The flask was heated and shaken for 15 min in a boiling water bath. After cooling, 1 mL of each sample was transferred to an Eppendorf tube and stored at −80 °C pending biochemistry analysis.

### 2.2. Analysis of Nitrate (NO_3_^−^) and Nitrite (NO_2_^−^)

Eppendorf tubes were thawed and centrifuged at 13,000 rpm for 10 min and at 4 °C. The supernatant was then collected and 10 µL of each sample was injected into a dedicated HPLC analyser (ENO-30; Eicom, San Diego, CA, USA) to measure NO_3_^−^ and NO_2_^−^. Briefly, NO_3_^−^ and NO_2_^−^ were separated on a reverse-phase column packed with polystyrene polymer (NO-PAK 4.6 × 50 mm, EICOM; Amuza, Inc., San Diego, CA, USA). NO_3_^−^ was reduced to NO_2_^−^ in a reduction column packed with copper-plated cadmium filings (NO-RED EICOM; Amuza, Inc.). NO_2_^−^ was mixed with a Griess reagent to form a purple azo dye in a reaction coil. The separation and reaction columns, along with the reaction coil, were placed in a column oven set at 35 °C. The absorbance of the dye at 540 nm was measured with a flow-through spectrophotometer (NOD-30; Eicom). 

The mobile phase (10% methanol, 0.15 M NaCl/NH_4_Cl, and 0.5 g/L 4Na-EDTA) and reactor phase (10% methanol, 1.25% HCl containing 5 g/L of sulfanilamide with 0.25 g/L of *N*-naphthylethylenediamine) were delivered at flow rates of 0.33 mL/min and 0.10 mL/min, respectively. A standard curve was produced by injecting 10 μL of water with sodium NO_3_^−^ (NaNO_3_^−^/7631-99-4; Sigma Aldrich, St. Louis, MO, USA) and sodium NO_2_^−^ (NaNO_2_^−^/7632-00-0; Sigma Aldrich) at different concentrations (1.95 μM, 62.5 μM, and 250 μM). The nori sample was diluted 1:100 using a carrier solution containing 10% methanol, 0.15 M NaCl/NH_4_Cl, and 0.5 g/L 4Na-EDTA. 

### 2.3. Analysis of Iodine

The extraction of iodine was performed according to Fecher et al. (1998) [25] using an alkali-based extraction to ensure volatile iodine species were kept in solution. Around 100 mg of each homogenised sample was added to separate acid-washed vials (*n* = 3/species, total = 69 samples). Procedural blanks (containing reagents only, *n* = 3) and a certified reference material (CRM; *n* = 3) with a known concentration of total iodine were also included to ensure the reliability and validity of measurements. Then, 5 mL of high-purity water and 1 mL of 25% tetramethylammonium hydroxide (TMAH) were added to each vial. The vials were sealed with Teflon-lined screw top caps and mixed before being placed in an oven at 90 ± 5 °C, where the samples were left for 3 h to digest. Following digestion, the samples were transferred to 15 mL centrifuge tubes and diluted to 10 mL with high-purity water. To each sample, Indium and Iridium were spiked in as internal standards to monitor instrument drift throughout the analysis, ready for quantification using inductively coupled plasma mass spectrometry (ICP-MS). A series of dissolved standards were analysed to compare with the unknown samples. The instrument detection limit was 0.0021 µg iodine/L, which equates to 0.19 mg iodine/kg. Analysis of the CRMs showed a concentration of 106.9 ± 5.0 mg/kg, which is similar to other reports (120 ± 4 mg/kg); [26], ensuring the validity and reliability of the measurement of the samples. 

### 2.4. Percentage Contribution to ADI for NO_3_^−^ and NO_2_^−^ Levels

The ADI for NO_3_^−^ and NO_2_^−^ is 3.7 mg/kg body mass/day [12] and 0.07 mg/kg body weight/day [19], respectively. Consequently, the percentage contribution to the ADI was calculated using the mean NO_3_^−^ and NO_2_^−^ concentrations obtained from the consumption of 5 g of edible algae. These calculations were based on the estimated ADI for a 70 kg adult (259 mg/day for NO_3_^−^ and 4.9 mg/day for NO_2_^−^).

### 2.5. Percentage Contribution to the UL for Iodine

The percentage contribution to the UL for iodine was calculated based on the iodine concentration obtained from consuming 5 g of edible algae. This calculation references the estimated UL of 600 µg/day of iodine [22]. 

### 2.6. Statistical Analyses

Data were presented as mean ± standard deviation. The normality of data was assessed using the Shapiro-Wilk test. Differences in the NO_3_^−^, NO_2_^−^, and iodine concentrations between different edible algae were compared using the Kruskal-Wallis H test. Differences in the NO_3_^−^, NO_2_^−^, and iodine content between macroalgae were assessed using a Mann–Whitney U test. The association between NO_3_^−^, NO_2_^−^, and iodine concentrations was analysed using two-tailed Spearman’s rank correlation analyses. Data were analysed using the statistical software SPSS (version 28). The level of significance was set at *p* < 0.05.

## 3. Results

The concentrations of NO_3_^−^, NO_2_^−^, and iodine are presented in Table 2. The coefficients of variation for the NO_3_^−^ and NO_2_^−^ analyses were 7.8 ± 7.5% and 10.1 ± 8.0%, respectively. Nori (N1) (17,191.33 ± 980.89 mg/kg), kombu powder (KO2) (4475.33 ± 42.36 mg/kg), and sweet kelp (SK1) (3183.00 ± 147.00 mg/kg) exhibited the highest concentrations of NO_3_^−^ (*p* < 0.001) (Table 2). Both *Spirulina* samples (S1 and S2) (7.02 ± 0.13 mg/kg and 6.26 ± 0.08 mg/kg, respectively) had the highest concentrations of NO_2_^−^ compared to other edible algae (*p* < 0.001). Regarding iodine, kelp (KE1) (6569.87 ± 412.26 mg/kg) had the highest concentration, followed by kombu (KO1) (4061.30 ± 271.66 mg/kg) and sweet kelp (SK1) (3882.07 ± 214.96 mg/kg), with significantly higher levels than the other samples (*p* < 0.001). A moderate, negative, and significant association was observed between NO_2_^−^ and iodine (r = −0.49; *p* = 0.018). NO_3_^−^ and iodine concentrations in macroalgae (NO_3_^−^: 4050.13 ± 1925.01 mg/kg; iodine: 1925.01 ± 1455.80 mg/kg) were significantly higher than in microalgae (NO_3_^−^: 55.73 ± 93.69 mg/kg; iodine: 17.61 ± 34.87 mg/kg; *p* < 0.001 for both).

Table 3 shows the contributions of NO_3_^−^ and NO_2_^−^ to the ADI, as well as the contribution of iodine to the UL for the analysed samples. The consumption of standard portion sizes of edible algae (5 g/day) did not exceed the ADI levels for either NO_3_^−^ or NO_2_^−^. However, the same standard portion size (5 g) of some edible algae surpassed the UL for iodine. Dulse (D2), pepper dulse fresh (PD1), wakame (W1), or bladderwrack (B1) exceeded the iodine UL by 132 and 250%. Even more significantly, kelp (KE1), kombu kelp (KK1), sweet kelp (SK1), kombu (KO1, KO2, and KO3), arame (A1), and egg wrack (EG1) exceeded the iodine UL by 625 and 5500%.

## 4. Discussion

The main finding of this study was the elevated NO_3_^−^ concentration in nori (N1) compared to the rest of the samples. The sweet kelp (SK1) and kombu powder (KO2) samples exhibited significant NO_3_^−^ levels, although these were nearly four times lower than in nori (N1). Additionally, the NO_3_^−^ and iodine content in edible macroalgae species was significantly higher than in microalgae species. 

These results align with a previous study showing that nori species marketed in Spain had the highest NO_3_^−^ concentration [15]. However, the NO_3_^−^ concentration reported in that study was nearly four times lower (3183 ± 2279 mg/kg) than in our study (>17,000 mg/kg). Several environmental factors can significantly influence the absorption of NO_3_^−^ by edible algae, including water temperature, sunlight exposure, water quality, and cultivation and harvesting practices [27]. Other factors that can affect the nutritional status of commercial edible algae include the origin, commercial status, and shelf life [28]. However, this study did not focus on these aspects. Our main goal was to investigate whether edible algae could be a natural source of NO_3_^−^. To achieve this, we used highly sensitive HPLC, which is considered one of the gold-standard approaches for measuring NO_3_^−^ and NO_2_^−^ in biological samples [16]. This was one of the main strengths of our study compared to previous research in this field [15]. 

We compared the NO_3_^−^ content in edible algae with that in terrestrial vegetables such as rocket, spinach, and lettuce, which are known to accumulate substantial amounts of NO_3_^−^. The European Commission has set NO_3_^−^ concentration limits of up to 5000 mg/kg in lettuce and 7000 mg/kg in rocket, as per Regulation (EC) No. 1258/2011, amending Regulation (EC) No. 1881/2006 [29,30]. These limits have been retained by the UK post-Brexit [31] and align with the ADI for NO_3_^−^ of 3.7 mg/kg body mass/day established by the EFSA [12], which is based on the potential association between dietary NO_3_^−^ intake and cancer risk [5]. However, recent evidence suggests that plant-based NO_3_^−^ may actually be protective against cancer and cardiovascular disease [14]. Therefore, a comprehensive understanding of NO_3_^−^’s role in human health is crucial to harness its potential benefits. 

To the best of our knowledge, there are no regulations limiting the amount of NO_3_^−^ in commercially available edible algae. Our results showed that nori can concentrate higher NO_3_^−^ levels compared to terrestrial vegetables, whereas sweet kelp and kombu powder samples contained amounts similar to those in lettuce and rocket [32]. However, the dietary portion size of edible algae is between 5–8 g/day (dry weight) [33], while the portion size for rocket and lettuce is around 80 g [34]. Consequently, the average NO_3_^−^ intake in a dietary portion of nori (5 g) could provide around 85 mg of NO_3_^−^, whereas a portion of rocket (80 g) could provide over 500 mg of NO_3_^−^. Thus, it is very unlikely that dietary consumption of edible algae within the recommended levels will exceed the ADI for NO_3_^−^. 

On the other hand, current research suggests that consuming at least 250 mg (4 mmol) of NO_3_^−^ from terrestrial vegetables can provide some physiological benefits, such as reduced blood pressure [4]. Based on our data, to achieve this amount, one would need to consume at least 15 g of nori (N1) and over 50 g of sweet kelp (SK1) or kombu powder (KO2) dry edible algae. However, some studies have reported a significant reduction in blood pressure levels with lower consumption of edible algae (2 g) [35,36], particularly with nori. Further research is needed to elucidate whether the antihypertensive effect of edible algae is least partially due to their NO_3_^−^ concentration. 

Regarding NO_2_^−^, our analysis revealed that microalgae samples, particularly *Spirulina* (S1), had the highest concentration (>7 mg/kg). This could be related to the presence of NO_3_^−^ reductase enzymes in these samples, which reduce NO_3_^−^ to NO_2_^−^ [37]. In comparison, some terrestrial vegetables, like Swiss chard and wild rocket, have been reported to contain NO_2_^−^ levels exceeding 50 mg/kg of NO_2_^−^, though such high levels are uncommon [38,39]. Most studies have reported NO_2_^−^ values below 2 mg/kg in vegetables like lettuce and rocket [38,39]. Consistent with these findings, we recently reported NO_2_^−^ values of approximately 1 mg/kg in fresh beetroot juice [40], which is seven times lower than the NO_2_^−^ concentration observed in *Spirulina* in this study. The ADI level for NO_2_^−^ is 0.07 mg/kg body weight/day, so nearly 5 mg/day for a standard 70 kg person. Consequently, consuming a standard portion size of *Spirulina* (5 g) would contribute significantly less (<0.04 mg) to this threshold. 

Edible algae are also an important dietary source of iodine. In our study, the iodine concentration observed in nori (N1), kombu (KK1), and wakame (W1) was consistent with previously reported values for edible algae [41]. We found that kelp (KE1) had the highest iodine concentration, followed by kombu (KO1) and sweet kelp (SK1). Specifically, kelp (KE1) had the highest iodine concentration (6569 mg/kg). Consuming just 9 g of this algae would provide the recommended daily intake of 150 µg/day, and 20 g would meet the upper intake level of 600 μg/day for adults [22]. However, excessive and recurrent iodine intake can adversely affect thyroid function [21]. It is also important to note that the bioavailability of iodine from edible algae has been reported to be around 50% [42,43,44]. Further research is needed to better understand the impact of edible algae consumption on thyroid health.

We did not observe an association between the NO_3_^−^ or NO_2_^−^ concentration and iodine levels in the samples we analysed. In fact, we found that nori (N1), the sample with the highest NO_3_^−^ concentration, had low iodine levels (16.3 mg/kg). To achieve the recommended daily intake of iodine, nearly 10 g of nori would be needed, and more than 35 g to reach the upper intake level [22]. This amount of nori would provide about 170 mg of NO_3_^−^ (2.7 mmol), which is below the ADI levels for NO_3_^−^, but also less than the amount of NO_3_^−^ (4 mmol) suggested to provide physiological benefits [4]. Additionally, previous research has suggested that salivary NO_3_^−^ uptake may compete with iodine [20]. However, this competition has not been demonstrated when iodine and NO_3_^−^ were administered together in healthy adults [45].

Concerns have been raised about the accumulation of toxic metals in edible algae, particularly in areas with industrial contamination or poor sewage systems [46]. Elements such as cadmium (Cd), lead (Pb), or mercury (Hg) can be present in marine waters as pollutants [46]. However, this risk is not unique to edible algae; terrestrial vegetables can also accumulate heavy metals when soils are contaminated [47]. Regarding NO_3_^−^ and NO_2_^−^, consuming small amounts of edible algae is unlikely to pose a higher risk of toxicity compared to terrestrial vegetables, except for some edible algae with high iodine concentrations. Nevertheless, monitoring the accumulation of harmful substances in marine vegetables is essential. 

While this study focused on the use of edible algae in the form of food or supplements (powder) in the human diet, it is important to note that algae can also offer solutions to urgent challenges such as water pollution. By absorbing excess nutrients, including nitrates, from aquatic environments, algae can help mitigate the harmful effects of nutrient runoff and eutrophication, thereby contributing to improved water quality and ecosystem health [48]. Additionally, algae with high NO_3_^−^ content can be utilised as organic fertilisers in agriculture, providing a sustainable alternative to chemical fertilisers and promoting soil health and productivity [49]. Thus, exploring the composition of algae is important for addressing environmental concerns and promoting sustainable practices across different sectors. 

This study had several limitations worth discussing. Like terrestrial vegetables, the NO_3_^−^ and NO_2_^−^ concentrations in edible algae may vary throughout the year due to environmental and harvesting conditions. Consequently, future studies should aim to analyse the variability in the NO_3_^−^ and NO_2_^−^ concentrations in these commercial products across different seasons. Additionally, we did not differentiate between food products and edible algae-based supplements (powder) in this study, as both can be used to enhance the composition of recipes. Furthermore, the bioavailability of NO_3_^−^, NO_2_^−^, and iodine was not analysed since our main aim was to identify edible algae with the highest content of these molecules. We aim to conduct further studies using similar products to analyse the bioavailability of NO_3_^−^, NO_2_^−^ and iodine in humans and to compare it with that of high-NO_3_^−^ terrestrial vegetables. Besides NO_3_^−^, NO_2_^−^, and iodine, future studies should also analyse other nutrients and investigate the presence of potentially harmful compounds such as heavy metals, as these are significant concerns regarding the nutritional value of edible algae. 

In summary, this study demonstrated that certain edible macroalgae species, particularly nori (N1) and kombu powder (KO2), can contain substantial quantities of NO_3_^−^, surpassing levels found in terrestrial vegetables like rocket. However, consuming the recommended portion size of commercial edible algae (5–8 g/day) is unlikely to exceed the NO_3_^−^ ADI levels. NO_2_^−^ was more abundant in edible microalgae species, especially *Spirulina* (S1 and S2). Nonetheless, similar to NO_3_^−^, consuming standard portion sizes is unlikely to exceed the NO_2_^−^ ADI levels. Macroalgae species such as kelp (KE1), sweet kelp (SK1), and kombu (KO1) exhibited higher iodine levels, and the consumption of small portions of these species (<1 g) could exceed the UL of iodine. We did not find an association between NO_3_^−^ and iodine levels, but we did find a negative association between NO_2_^−^ concentration and iodine. Thus, edible algae rich in NO_3_^−^ or NO_2_^−^ did not necessarily contain large quantities of iodine. These findings are relevant to nutritionists, researchers, and commercial companies interested in the nutritional properties of edible algae. 

## Figures and Tables

**Table 1 foods-13-02615-t001:** Edible algae analysed in this study.

Package Label	Species	Colour	Type	Origin on the Label	Habitat	Date of Packaging	Best Before Date	Sample ID
100% organic dulse (dried), The Cornish Seaweed Company	*Palmaria palmata*	Red	Macroalgae	Cornwall, United Kingdom	Marine	Unknown	05.24	D1
100% pure dulse flakes (dried), Mara seaweed	*Palmaria palmata*	Red	Macroalgae	Scotland	Marine (hand-harvested from pure cold waters)	06.06.22	11.24	D2
Pepper dulse (fresh), The Irish Seaweed Company	*Osmundea pinnatifida*	Red	Macroalgae	Unknown	Marine	Unknown	Unknown	PD1
Japanese nori (dried), Clearspring	*Porphyra yezoensis*	Red	Macroalgae	Japan	Marine (cultivated in the sheltered bays around the Japanese coastline)	17.12.21	16.12.23	N1
100% organic sea spaghetti (dried), The Cornish Seaweed Company	*Himanthalia elongata*	Brown	Macroalgae	Cornwall, United Kingdom	Marine	Unknown	06.24	SS1
Organic sea spaghetti (dried), Porto-Muiños Las Verduras del Mar	*Himanthalia elongata*	Brown	Macroalgae	Galician coast (Bay of Biscay and Portuguese waters), Spain	Marine (hand-harvested from its natural environment)	28.05.21	05.24	SS2
100% organic kelp (dried), The Cornish Seaweed Company	*Laminaria digitata*	Brown	Macroalgae	Cornwall, United Kingdom	Marine	Unknown	11.23	KE1
Kombu kelp (dried), The Irish Seaweed Company	*Laminaria digitata*	Brown	Macroalgae	Unknown	Marine	Unknown	30.11.22	KK1
Sweet kelp (Kombu Royale) (dried), The Irish Seaweed Company	*Saccharina latissimi*	Brown	Macroalgae	Northern Ireland	Marine	Unknown	30.11.22	SK1
Organic wakame (dried), Porto-Muiños Las Verduras del Mar	*Undaria pinnatifida*	Brown	Macroalgae	Spain	Marine	26.05.21	05.24	W1
Organic Atlantic wakame (dried), Clearspring	*Undaria pinnatifida*	Brown	Macroalgae	European	Marine	21.12.21	31.12.23	W2
Kombu (dried), Mara seaweed	*Laminaria digitata*	Brown	Macroalgae	Scotland	Marine (hand-harvested from pure cold water)	06.06.22	11.24	KO1
Kombu powder, 100% brown kelp (dried, powder), Mara seaweed	*Laminaria digitata*	Brown	Macroalgae	Scotland and Ireland	Marine	30.05.22	10.24	KO2
Japanese Hokkaido kombu (dried), Clearspring	*Laminaria japonica*	Brown	Macroalgae	Japan	Marine (harvested in the luscious waters of the island of Hokkaido)	24.01.22	24.02.25	KO3
Japanese arame (dried), Clearspring	*Eisenia bicyclis*	Brown	Macroalgae	Japan	Marine	16.06.21	30.06.23	A1
Egg wrack (dried), The Irish Seaweed Company	*Ascophyllum nodosum*	Brown	Macroalgae	Northern Ireland	Marine	Unknown	30.11.22	EG1
Bladderwrack (dried), The Irish Seaweed Company	*Fucus vesiculosus*	Brown	Macroalgae	Northern Ireland	Marine	Unknown	30.11.22	B1
100% organic ocean greens (dried), The Cornish Seaweed Company	*Ulva* sp.	Green	Macroalgae	Cornwall, United Kingdom	Marine	Unknown	06.24	U1
*Dunaliella salina* (powder), Green Heart Store	*Dunaliella salina*	Green	Microalgae	Unknown	Cultivated in water with a very high salt content	Unknown	04.23	DS1
Organic *Spirulina* (powder), BuyWholefoodsOnline	*Arthrospira platensis*	Blue-green	Microalgae	China		16.06.22	21.06.23	S1
Organic *Spirulina* (powder), Naturya	*Arthrospira platensis*	Blue-green	Microalgae	Non-European	Freshwater pools under warm sunlight	Unknown	01.24	S2
Organic *Chlorella* (powder cracked cell wall), Health Essentials Direct	*Chlorella*	Green	Microalgae	Non-European	Freshwater *	18.02.20	03.24	C1
Organic *Chlorella* (powder), BuyWholefoodsOnline	*Chlorella*	Green	Microalgae	China	Freshwater *	20.04.22	22.05.25	C2

“Date of packaging” and “Best before date” are expressed as DD.MM.YY or MM.YY, when available. * Searched online as it was not mentioned on the package.

**Table 2 foods-13-02615-t002:** Concentrations of nitrate (NO_3_^−^), nitrite (NO_2_^−^), and iodine in commercial edible algae.

	Sample ID	NO_3_^−^ (mg/kg)	NO_2_^−^ (mg/kg)	Iodine (mg/kg)
Macroalgae				
Dulse (The Cornish Seaweed Company)	D1	1303.33 ± 8.50 ^a^	0.86 ± 0.17 ^b^	61.03 ± 13.18 ^c^
Dulse (Mara seaweed)	D2	1069.33 ± 5.77 ^d^	0.47 ± 0.12 ^b^	158.53 ± 5.16 ^c^
Pepper dulse fresh (The Irish Seaweed Company)	PD1	185.00 ± 1.73 ^e^	0.00 ± 0.00 ^f^	297.97 ± 25.01 ^c^
Nori (Clearspring)	N1	17,191.33 ± 980.89 ^g^	3.64 ± 2.38 ^g^	16.30 ± 0.56 ^c^
Sea spaghetti (The Cornish Seaweed Company)	SS1	57.33 ± 3.21 ^e^	0.32 ± 0.07 ^b^	74.40 ± 11.65 ^c^
Sea spaghetti (Porto-Muiños Las Verduras del Mar)	SS2	111.33 ± 13.20 ^e^	0.39 ± 0.02 ^b^	63.90 ± 4.93 ^c^
Kelp (The Cornish Seaweed Company)	KE1	858.67 ± 19.73 ^h^	0.34 ± 0.01 ^b^	6569.87 ± 412.26 ^g^
Kombu kelp (The Irish Seaweed Company)	KK1	197.00 ± 1.73 ^e^	0.27 ± 0.05 ^b^	1978.93 ± 211.75 ^i^
Sweet kelp (Kombu Royale) (The Irish Seaweed Company)	SK1	3183.00 ± 147.00 ^g^	0.48 ± 0.02 ^b^	3882.07 ± 214.96 ^j^
Wakame (Porto-Muiños Las Verduras del Mar)	W1	513.67 ± 21.29 ^k^	1.55 ± 0.17 ^b^	160.17 ± 1.76 ^c^
Wakame (Clearspring)	W2	178.67 ± 4.04 ^e^	0.68 ± 0.02 ^b^	99.77 ± 2.78 ^c^
Kombu (Mara seaweed)	KO1	188.00 ± 1.73 ^e^	0.40 ± 0.06 ^b^	4061.30 ± 271.66 ^l^
Kombu powder (Mara seaweed)	KO2	4475.33 ± 42.36 ^g^	0.79 ± 0.06 ^b^	3261.70 ± 110.65 ^m^
Kombu (Clearspring)	KO3	7.33 ± 2.52 ^e^	0.22 ± 0.03 ^b^	2933.37 ± 154.19 ^n^
Arame (Clearspring)	A1	89.67 ± 16.07 ^e^	0.49 ± 0.01 ^b^	750.87 ± 41.83 ^g^
Egg wrack (The Irish Seaweed Company)	EG1	401.00 ± 29.72 ^o^	0.34 ± 0.12 ^b^	1579.60 ± 222.63 ^p^
Bladderwrack (The Irish Seaweed Company)	B1	88.33 ± 7.09 ^e^	0.38 ± 0.07 ^b^	165.83 ± 7.10 ^c^
*Ulva* sp. (The Cornish Seaweed Company)	U1	220.00 ± 27.50 ^o^	0.33 ± 0.10 ^b^	88.87 ± 0.99 ^c^
Microalgae				
*Dunaliella salina* (Green Heart Store)	DS1	223.00 ± 21.93 ^o^	0.35 ± 0.06 ^b^	79.97 ± 0.76 ^c^
*Spirulina* powder (BuyWholefoodsOnline)	S1	23.00 ± 4.00 ^e^	7.02 ± 0.13 ^q^	1.63 ± 0.06 ^c^
*Spirulina* powder (Naturya)	S2	8.33 ± 3.06 ^e^	6.26 ± 0.08 ^r^	1.07 ± 0.12 ^c^
*Chlorella* powder (Health Essentials Direct)	C1	15.33 ± 3.79 ^e^	1.61 ± 0.07 ^b^	2.20 ± 0.50 ^c^
*Chlorella* powder (BuyWholefoodsOnline)	C2	9.00 ± 2.00 ^e^	0.40 ± 0.06 ^b^	3.17 ± 3.03 ^c^

Values are mean ± standard deviations. Abbreviations: NO_3_^−^, nitrate; NO_2_^−^, nitrite. ^a^ Significantly different (*p* < 0.05) compared to the other edible algae, except D2 and KE1. ^b^ Significantly different (*p* < 0.05) compared to N1, S1, and S2. ^c^ Significantly different (*p* < 0.05) compared to KE1, KK1, SK1, KO1, KO2, KO3, A1, and EG1. ^d^ Significantly different (*p* < 0.05) compared to the other edible algae, except D1, KE1, and W1. ^e^ Significantly different (*p* < 0.05) compared to D1, D2, N1, KE1, SK1, and KO2. ^f^ Significantly different (*p* < 0.05) compared to N1, S1, S2, and C1. ^g^ Significantly different (*p* < 0.05) compared to the other edible algae. ^h^ Significantly different (*p* < 0.05) compared to the other edible algae, except D1, D2, U1, EG1, W1, and DS1. ^i^ Significantly different (*p* < 0.05) compared to the other edible algae, except EG1. ^j^ Significantly different (*p* < 0.05) compared to the other edible algae, except KO1. ^k^ Significantly different (*p* < 0.05) compared to D1, N1, SW1, and KO2. ^l^ Significantly different (*p* < 0.05) compared to the other edible algae, except SK1. ^m^ Significantly different (*p* < 0.05) compared to the other edible algae, except KO3. ^n^ Significantly different (*p* < 0.05) compared to the other edible algae, except KO2. ^o^ Significantly different (*p* < 0.05) compared to D1, D2, N1, SK1, and KO2. ^p^ Significantly different (*p* < 0.05) compared to the other edible algae, except KK1. ^q^ Significantly different (*p* < 0.05) compared to the other edible algae, except S2. ^r^ Significantly different (*p* < 0.05) compared to the other edible algae, except S1.

**Table 3 foods-13-02615-t003:** The percentage (%) contribution to the acceptable daily intake (ADI) for both nitrate (NO_3_^−^) and nitrite (NO_2_^−^), and the percentage (%) contribution to the tolerable upper intake level (UL) for iodine in commercial edible algae.

	Sample ID	NO_3_^−^ % Contribution to ADI (5 g Serving) †	NO_2_^−^ % Contribution to ADI (5 g Serving) †	Iodine % Contribution to UL (5 g Serving) ‡
Macroalgae				
Dulse (The Cornish Seaweed Company)	D1	2.52 ± 0.02	0.09 ± 0.02	50.86 ± 10.98
Dulse (Mara seaweed)	D2	2.06 ± 0.01	0.05 ± 0.01	132.11 ± 4.30
Pepper dulse (The Irish Seaweed Company)	PD1	0.36 ± 0.00	0.00 ± 0.00	248.31 ± 20.84
Nori (Clearspring)	N1	33.19 ± 1.89	0.37 ± 0.24	13.58 ± 0.46
Sea spaghetti (The Cornish Seaweed Company)	SS1	0.11 ± 0.01	0.03 ± 0.01	62.00 ± 9.71
Sea spaghetti (Porto-Muiños Las Verduras del Mar)	SS2	0.21 ± 0.03	0.04 ± 0.00	53.25 ± 4.11
Kelp (The Cornish Seaweed Company)	KE1	1.66 ± 0.04	0.03 ± 0.00	5474.89 ± 343.55
Kombu kelp (The Irish Seaweed Company)	KK1	0.38 ± 0.00	0.03 ± 0.00	1649.11 ± 176.46
Sweet kelp (Kombu Royale) (The Irish Seaweed Company)	SK1	6.14 ± 0.28	0.05 ± 0.00	3235.06 ± 179.13
Wakame (Porto-Muiños Las Verduras del Mar)	W1	0.99 ± 0.04	0.16 ± 0.02	133.47 ± 1.47
Wakame (Clearspring)	W2	0.34 ± 0.01	0.07 ± 0.00	83.14 ± 2.32
Kombu (Mara seaweed)	KO1	0.36 ± 0.00	0.04 ± 0.01	3382.42 ± 226.38
Kombu powder (Mara seaweed)	KO2	8.64 ± 0.08	0.08 ± 0.01	2718.08 ± 92.21
Kombu (Clearspring)	KO3	0.01 ± 0.00	0.02 ± 0.00	2444.47 ± 128.49
Arame (Clearspring)	A1	0.17 ± 0.03	0.05 ± 0.00	625.72 ± 34.86
Egg wrack (The Irish Seaweed Company)	EG1	0.77 ± 0.06	0.03 ± 0.01	1316.33 ± 185.52
Bladderwrack (The Irish Seaweed Company)	B1	0.17 ± 0.01	0.04 ± 0.01	138.19 ± 5.92
*Ulva* sp. (The Cornish Seaweed Company)	U1	0.42 ± 0.05	0.03 ± 0.01	74.06 ± 0.82
Microalgae				
*Dunaliella salina* (Green Heart Store)	DS1	0.43 ± 0.04	0.04 ± 0.01	66.64 ± 0.63
*Spirulina* powder (BuyWholefoodsOnline)	S1	0.04 ± 0.01	0.72 ± 0.01	1.36 ± 0.05
*Spirulina* powder (Naturya)	S2	0.02 ± 0.01	0.64 ± 0.01	0.89 ± 0.10
*Chlorella* powder (Health Essentials Direct)	C1	0.03 ± 0.01	0.16 ± 0.01	1.83 ± 0.42
*Chlorella* powder (BuyWholefoodsOnline)	C2	0.02 ± 0.00	0.04 ± 0.01	2.64 ± 2.52

Values are mean ± standard deviations. Abbreviations: NO_3_^−^, nitrate; NO_2_^−^, nitrite; ADI, acceptable daily intake; UL, tolerable upper intake level. † The % contribution to the ADI was calculated using the mean NO_3_^−^ and NO_2_^−^ concentrations obtained from the consumption of 5 g of edible algae based on the estimated ADI for a 70 kg adult (259 mg/day for NO_3_^−^ and 4.9 mg/day for NO_2_^−^). ‡ The % contribution to the UL for iodine was calculated based on the iodine concentration obtained from consuming 5 g of edible algae, with reference to the estimated UL of 600 µg/day of iodine.

## Data Availability

The data that support the findings of this study are restricted for research use only. The data are not publicly available. Data are available from the authors upon reasonable request and with permission from the School of Health Professions, University of Plymouth at Plymouth, United Kingdom.

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
