# Peer review of "Nitrate, Nitrite, and Iodine Concentrations in Commercial Edible Algae: An Observational Study"

_foods, 2024, doi:10.3390/foods13162615_

Round 1

Reviewer 1 Report

Comments and Suggestions for Authors

This is a very ordinary account of the concentration of nitrate, nitrite and iodine from various 'processed macro- and micro-algal products destined to be consumed as food rather than as supplements. This aspect needs some clarification. Are the algal products food products and marketed as such or are these food supplements? Second, more detailed information about these products could or should be provided. The date of purchase is essentially meaningless. Why not the date of packaging, or date of production and how it was produced? For the latter origin as place, while acceptable, does not inform about source, e.g. freshwater body or cultivated or marine. Naturally produced or farmed. Some detail in this regard could be useful in assessing the content of the selected nutrients. Last, even though an alga may appear rich in a particular nutrient, how does this relate to nutrient availability? In conclusion, a more thought-provoking account of the value of the study and a more detailed analysis of the limitations of the results would improve the overall quality of this manuscript.

Minor comments

1. Species names such as Dunaliella and Chlorella should be upper case and italicised.

2. The statement "This is essential for photosynthesis, enabling these organisms to concentrate high levels of NO3 - [4]" should be carefully checked for accuracy/interpretation.

3. Please insert 'extracts' after "2.2. Preparation of edible algae"

Comments on the Quality of English Language

Some minor English language editing is required.

Author Response

Comments and Suggestions for Authors

This is a very ordinary account of the concentration of nitrate, nitrite and iodine from various 'processed macro- and micro-algal products destined to be consumed as food rather than as supplements. This aspect needs some clarification. Are the algal products food products and marketed as such or are these food supplements? Second, more detailed information about these products could or should be provided. The date of purchase is essentially meaningless. Why not the date of packaging, or date of production and how it was produced? For the latter origin as place, while acceptable, does not inform about source, e.g. freshwater body or cultivated or marine. Naturally produced or farmed. Some detail in this regard could be useful in assessing the content of the selected nutrients. Last, even though an alga may appear rich in a particular nutrient, how does this relate to nutrient availability? In conclusion, a more thought-provoking account of the value of the study and a more detailed analysis of the limitations of the results would improve the overall quality of this manuscript.

Response:

We appreciate the reviewer’s comments.

In this study, we aimed to analyse whether commercially available edible algae in the UK, in the form of food or supplements (powder), are a natural source of nitrate, nitrite, and iodine.

Research into dietary nitrate has significantly increased over the last decade, driven by its potential to enhance nitric oxide bioavailability. Recent evidence also suggests that dietary nitrate has a prebiotic effect on the oral microbiome. While most studies have focused on terrestrial vegetables, such as green leafy vegetables and beetroot, as natural sources of nitrate, algae also represent a promising yet underexplored option. Algae have the capacity to fix nitrate from seawater, potentially providing an additional valuable source of dietary nitrate.

We have included the date of packaging, best-before date, and habitat (including the production method) where the edible algae were produced in the new version of Table 1, when this information was available on the package. If not available, the information was sourced online and marked with an *. The packaging often lacks explicit details on whether the edible algae are naturally produced or farmed. However, by visiting the manufacturers' websites, we identified that some companies operate farms. Nevertheless, it is not always clear if all the edible algae are farmed, as some packages specify that the algae are hand-harvested from their natural environment.

We agree with the comment regarding nutrient availability, and this has been highlighted in the study’s limitations. This study was necessary to identify algae species with the highest nitrate content. We aim to conduct further studies using those species with the highest nitrate content to analyse nitrate availability and its physiological role in humans, and to compare these effects with those of land vegetables, such as lettuce. 

Minor comments

Response: Thank you for these comments. We have incorporated them into the revised manuscript.

  1. Species names such as Dunaliella and Chlorella should be upper case and italicised.

Response: Done as suggested.

  1. The statement "This is essential for photosynthesis, enabling these organisms to concentrate high levels of NO3 - [4]" should be carefully checked for accuracy/interpretation.

Response: This statement has been deleted in the revised version of the manuscript as we have improved the introduction.

  1. Please insert 'extracts' after "2.2. Preparation of edible algae"

Response: Done as suggested.

Comments on the Quality of English Language

Some minor English language editing is required.

Response: We appreciate this comment. We have addressed some editing issues in the revised manuscript, particularly in the introduction and discussion sections.

Reviewer 2 Report

Comments and Suggestions for Authors

The nutritional value and efficacy of edible algae have always been of great concern to scientists and society, but there is a lack of in-depth research on their nutritional and efficacy analysis, which also limits the deep development of edible algae. The author analyzed the nutritional components of various commercial edible algae, which is a very meaningful study. 

The following comments, hoping to help the author enhance the value and influence of this study.

1. Why didn't the author analyze other nutrients, such as organic matter, besides the rich nutritional content of algae?

2. The representativeness of nitrate and nitrite for nitrogen-containing nutrients is questionable, and the determination of compounds with nutritional effects may be more representative than the measurement of broad degradation products.

3. Is it meaningful or simpler to directly measure the elemental content of carbon, nitrogen, and iodine?

4. Lines 98-100, The origin, commercial status, and shelf life of these commercial algae can all affect the nutritional content of edible algae. The author needs to consider these influencing factors and indicate them in the article.

Comments on the Quality of English Language

No

Author Response

Comments and Suggestions for Authors

The nutritional value and efficacy of edible algae have always been of great concern to scientists and society, but there is a lack of in-depth research on their nutritional and efficacy analysis, which also limits the deep development of edible algae. The author analyzed the nutritional components of various commercial edible algae, which is a very meaningful study. 

Response:

We thank the reviewer for their comment.

The following comments, hoping to help the author enhance the value and influence of this study.

  1. Why didn't the author analyze other nutrients, such as organic matter, besides the rich nutritional content of algae?

Response:

We focused on nitrate and nitrite due to the growing interest in dietary sources rich in these anions over the past decade, driven by their potential health benefits. Research into dietary nitrate has significantly increased, aiming to enhance nitric oxide bioavailability. Recent evidence suggests that dietary nitrate may also have a prebiotic effect on the oral microbiome. While most studies have concentrated on terrestrial vegetables, such as green leafy vegetables and beetroot, as natural sources of nitrate, algae represent a promising yet underexplored alternative. Algae have the ability to fix nitrate from seawater, potentially providing an additional valuable source of dietary nitrate.

Additionally, we analysed iodine because edible algae are a natural source of it. Some commercial brands recommend consuming no more than 5 g/day of edible algae to reduce the risk of hyperthyroidism. This recommendation can also limit nitrate/nitrite intake from edible algae, thereby preventing the consumption of nitrate amounts that are associated with positive health effects (300 mg).

In future studies, we aim to analyse other compounds, such as ions and heavy metals. We have highlighted this objective in the limitations section of the discussion.

  1. The representativeness of nitrate and nitrite for nitrogen-containing nutrients is questionable, and the determination of compounds with nutritional effects may be more representative than the measurement of broad degradation products.

Response:

Inorganic nitrate from land plants, such as lettuce, can play important role in enhancing nitric oxide bioavailability in humans (Ashworth and Bescos, 2017; Bondonno et al. 2023; Pinaffi-Langley et al. 2023). We believe that inorganic nitrate from other natural sources, like edible algae, might have similar physiological effects. This study was necessary to identify algae species with the highest nitrate content. We aim to conduct further studies using those species with the highest nitrate content to analyse nitrate availability and its physiological role in humans, and to compare these effects with those of land vegetables, such as lettuce. 

References:

Ashworth, A. and R. Bescos, Dietary nitrate and blood pressure: evolution of a new nutrient? Nutrition research reviews, 2017. 30(2): p. 208-219.

Bondonno, C.P., et al., Nitrate: The Dr. Jekyll and Mr. Hyde of human health? Trends in Food Science & Technology, 2023.

Pinaffi-Langley, A.C.d.C., et al., Perspective: dietary nitrate from plant foods: a conditionally essential nutrient for cardiovascular health. Advances in Nutrition, 2023: p. 100158.

  1. Is it meaningful or simpler to directly measure the elemental content of carbon, nitrogen, and iodine?

Response:

We believe this is relevant as inorganic nitrate, nitrite and iodine could play important biological roles in the human body.

  1. Lines 98-100, The origin, commercial status, and shelf life of these commercial algae can all affect the nutritional content of edible algae. The author needs to consider these influencing factors and indicate them in the article.

Response:

We appreciate this comment, and this information has been added in the discussion of the revised manuscript.

‘Other factors that can affect the nutritional status of commercial edible algae are the origin, commercial status and shelf life(page 14, lines 14-15).